# Impact of Digital Twins and Metaverse on Cities: History, Current Situation, and Application Perspectives

**Zhihan Lv [1,*], Wen-Long Shang [2,3,4] and Mohsen Guizani [5]**

[1] Department of Game Design, Faculty of Arts, Uppsala University, SE-75105 Uppsala, Sweden
[2] Beijing Key Laboratory of Traffic Engineering, College of Metropolitan Transportation, Beijing University of Technology, Beijing 100124, China
[3] School of Traffic and Transportation, Beijing Jiaotong University, Beijing 100008, China
[4] Centre for Transport Studies, Imperial College London, London SW7 2AZ, UK
[5] Machine Learning Department, Mohamed Bin Zayed University of Artificial Intelligence (MBZUAI), Abu Dhabi P.O. Box 54115, United Arab Emirates
[*] Correspondence: lvzhihan@gmail.com

**Abstract:** To promote the expansion and adoption of Digital Twins (DTs) in Smart Cities (SCs), a detailed review of the impact of DTs and digitalization on cities is made to assess the progression of cities and standardization of their management mode. Combined with the technical elements of DTs, the coupling effect of DTs technology and urban construction and the internal logic of DTs technology embedded in urban construction are discussed. Relevant literature covering the full range of DTs technologies and their applications is collected, evaluated, and collated, relevant studies are concatenated, and relevant accepted conclusions are summarized by modules. First, the historical process and construction content of a Digital City (DC) under modern demand are analyzed, and the main ideas of a DC design and construction are discussed in combination with the key technology of DTs. Then, the metaverse is the product of the combination of various technologies in different scenes. It is a key component to promote the integration of the real world and the digital world and can provide more advanced technical support in the construction of the DC. DTs urban technology architecture is composed of an infrastructure terminal information center terminal and application server end. Urban intelligent management is realized through physical urban data collection, transmission, processing, and digital urban visualization. The construction of DTs urban platform can improve the city's perception and decision-making ability and bring a broader vision for future planning and progression. The interactive experience of the virtual world covered by the metaverse can effectively support and promote the integration of the virtual and real, and will also greatly promote the construction of SCs. In summary, this work is of important reference value for the overall development and practical adoption of DTs cities, which improves the overall operation efficiency and the governance level of cities.

**Keywords:** digital twins; digital city; smart city; Internet of things; intelligent urban management

## 1. Introduction

### 1.1. Research Background

Following the agricultural and industrial economies, the digital economy has become the main economic form in the last decade. With its rapid progression, extensive radiation, and unprecedented influence, the digital economy is promoting profound changes in the mode of production, way of life, and governance. It has become a key force in restructuring global factor resources, reshaping the global economic structure, and changing global competition patterns [1–3]. The digital economy is becoming a key force in restructuring global factor resources with its rapid progression, wide range of radiation, and unprecedented depth of influence, reshaping global economic structure and changing global competition patterns. In recent years, digitization has rapidly changed the form of

economic and social progression. At the same time, global instability factors are increasing, posing more challenges to urban governance, public services, and economic resilience. Ashraf (2021) [4] analyzed the impact of intelligent transportation, intelligent energy, intelligent infrastructure, intelligent health, intelligent agriculture, intelligent leisure, and other services on smart and traditional cities through an analytic hierarchy process and affirmed the application value of deep learning and Internet of Things in the construction of SCs. After understanding the security requirements of Internet of Things communication, Feng et al. (2021) [5] designed the interference source positioning scheme based on mobile trackers in the process of Internet of Things communication to reduce the interference attack communication network in the Internet of Things, so as to improve the ability of Internet of Things network to resist interference attack and ensure the communication security of SC. It has become a common practice to promote the comprehensive transformation of a city and break through the obstacles of economic and social operation. Cities are the main battlefield of economic construction, with the fastest progression, most abundant information, and highly concentrated capital [6]. The process of urban informatization has advanced rapidly, and the demand for geographic information is the most intense. However, in the process of development, cities will inevitably encounter practical problems and even become obstacles during the informatization progress. For example, the emergency response should be integrated with information to support decision-making, but the reality is that the spatial base is fragmented and the integration process takes time, making the achieved results unreliable. Therefore, establishing a continuous, unified, authoritative, and unique spatial foundation or geospatial framework in cities is an urgent task to support national economic informatization and local economic and social development [7–9].

With the progression of connectivity technologies such as AI and IoT, intelligent earth has been proposed, and hence the concept of Smart Cities (SCs) has been born [10]. Although the construction paths of SCs in different regions have different emphases, most of them have experienced a transition and iterative process from infrastructure-driven to process-driven and then to space-driven. Data-driven new industrial ecology has emerged in various places. The comprehensive development of the digital industry and the use of data as new energy is an important means to change the dependence of economic growth on natural energy. The combination of digitalization with energy-saving and low-carbon technologies is the focus, which can effectively promote the rational utilization of urban resources [11,12]. Digital Twins (DTs) cities make full use of government networks, data exchange platforms, and modern information technology to give full play to the monitoring and screening ability, business operation guidance ability, information transparency, collaborative support, and decision support of information systems. Thus, it can realize the rapid discovery, accurate positioning, and intelligent decision-making of urban problems. The efficiency of management and allocation of urban public resources have improved, and decision-makers with a tool to perceive the overall situation of a city and a platform to command and dispatch are provided. DTs originated from the industrialization of complex product development and is moving towards urbanization and globalization. The metaverse, which originated from the game and entertainment industry of building relationships, is moving from globalization to urbanization and industrialization. Chen et al. (2022) [13] believe that data were at the core of DTs, and the speed of data acquisition directly affected the interaction frequency of the virtual and real space of DTs. In the DTs and metaverse, data is its core supporting resource, and data is also the carrier and manifestation of information and knowledge. From the perspective of interaction, metaverse focuses on augmented reality technology to build a virtual world beyond the real world and to obtain an immersive user experience, while DTs focus more on virtual reality technology to generate the image of the real world.

Geographic information actually plays a key role in a Digital City's (DC) construction [14,15]. The Bureau of Surveying, Mapping, and Geographic Information made the pilot construction of the geospatial framework of a DC, which developed into an SC and turned into an industrial platform, now called a DTs city. In terms of technology, the

rapid development and transformation of new information technologies such as 5G, cloud computing, and blockchain also affect the construction process of DTs cities [16]. The city-based digital space enables the collection, cleaning, storage, and standardization of data through the physical, information, social, and commercial infrastructure of the city. Smart transport and wisdom cities are empowered through the fusion data model of urban traffic intelligent prediction and decision-making [17–19]. The DTs urban technology architecture is composed of the infrastructure, information center, and application server ends. Urban intelligent management can be realized through physical urban data collection, transmission, processing, and digital urban visualization. The infrastructure terminal is mainly responsible for the collection and transmission of urban data, the information center is responsible for data reception, processing, and transmission, and the application server is the practical adoption of the DTs city model [20,21].

### 1.2. Research Methods, Procedures, and Significance

DTs are no longer just a technology but a new mode of progression, a new path of transformation, and have a new power to promote profound changes in all industries. The metaverse is virtual reality, which emphasizes visual immersion, displaying a rich imagination and a fully virtual world. However, DTs is a copy of the only physical element in the real world; it is object-oriented and emphasizes physical reality. Connecting the DTs to the metaverse will provide all-round technical support for the construction of SCs. Facing the development requirements and functions of DTs, it is necessary to establish a more systematic DTs city, which requires a very complex system engineering.

This work mainly focuses on urban development under the background of DTs and digital technology. In the second section, the historical process and construction concept of DC under the requirements of modernization are clarified, and the management mode of DC is analyzed. In the third section, based on the characteristics of DTs technology, the key technologies in the construction of SC and the role of DTs technology are explored. In the fourth section, combined with the development process of modern SC, the development prospect and technological breakthroughs of digital SC are comprehensively planned. By summarizing the current status of digital SC in the development process, the development trend is prospected and analyzed to provide a reference for the subsequent urban construction and management. The construction of a SC based on DTs improves the management and allocation efficiency of urban public resources and provides decision-makers with a global situation awareness tool and a command and scheduling platform.

## 2. DC Construction under the Demand of Modernization

### 2.1. Goals and Ideas of DC

In his speech of January 1998, US Vice President Al Gore first proposed the concept of "digital earth". Gore suggested that we need a "digital earth"—a virtual earth based on earth coordinates embedded with massive geographical data, with multiresolution and three-dimensional visualization [22–24]. A DC is an important part of "digital earth" and a concrete embodiment of "digital earth" in cities. Cheng et al. (2022) [25] established an optimization model for energy-saving traffic scheduling problems. Based on an energy-saving model, the traffic prediction algorithm and the multilayer virtual topology traffic scheduling algorithm were solved, and the final energy-saving scheduling scheme was obtained. The work played a positive role in reducing the energy consumption of data center network and further provided a reference for the development direction of Internet of Things technology and SC. The development of the "digital earth" has gone through three stages, namely digitization, informatization, and intelligence. Digitization is an early stage, but data has not been effectively classified and managed, so it cannot be called information. Information theory refers to the meaningful content of data as information. In the information stage, data can be effectively classified, stored, and managed and becomes an effective resource. Novak et al. [26] suggested that urban informatization is developing toward intelligence with the emergence and adoption of new connectivity technologies

such as sensor networks. The evolution and development of human society from "physical age" to "material age" and then to "intellectual age" is the general trend of the continuous upgrading of civilization. When production tools progress from agricultural equipment to industrial equipment, information equipment, and intelligent equipment, society will progress from an agricultural society to an industrial society, information society, and network society. Cities are also changing from agricultural towns, industrial cities, and DCs to SCs. Therefore, SC becomes the inevitable direction of urban progression.

Valdenebro and Gimena [27] discussed that cities are facing numerous challenges that threaten their long-term sustainability. These challenges can affect cities' economies, businesses, and residents, involving core infrastructures such as transport, water resources, energy, and communications. These challenges must be comprehensively addressed if cities are to build and maintain sustainable urban environments. DC is far more than a highly integrated set of technologies. For cities to evolve, they must utilize and optimize their economic capabilities, physical assets, culture, political will, technology, and business environment [28,29]. The comparison of operation modes between traditional cities and DC is shown in Figure 1.

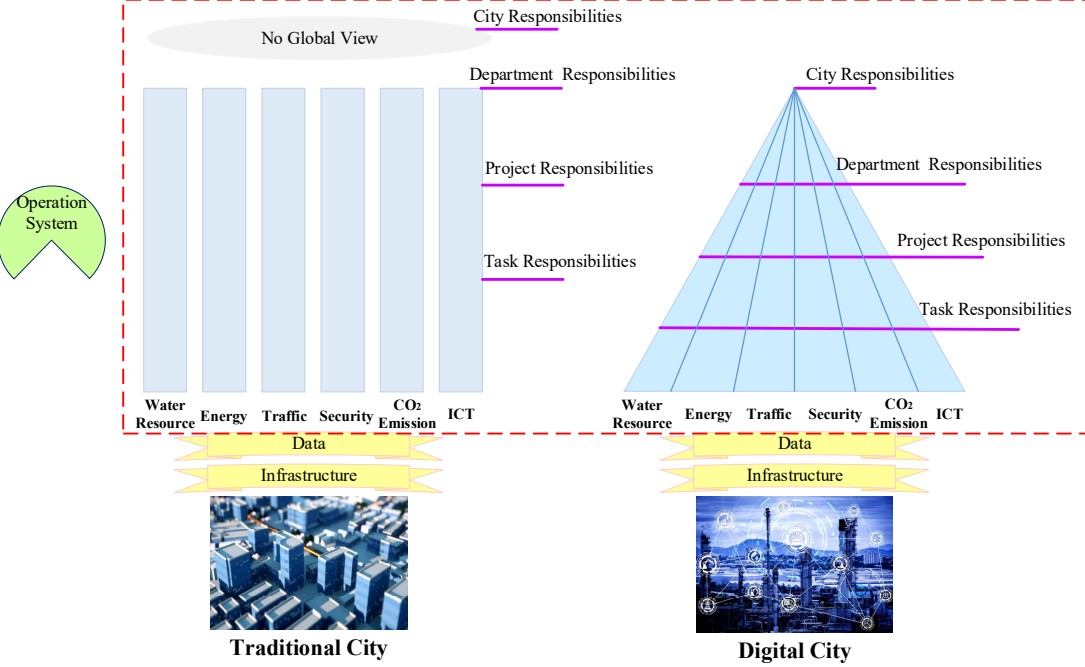

**Figure 1.** Comparison of operation modes between traditional cities and DCs.

The development of global DC has roughly gone through four stages. The first stage is the construction of the network infrastructure; the second stage is the construction of the municipal government and enterprise internal information system; the third stage is the interconnection between the municipal government and the upstream and downstream enterprises through the internet; and the fourth stage is the formation of network society, network community, and DC. China's DC has gone through more than 20 years since it was proposed in 1999, but the development process is not exactly the same as that of European and American countries [30–32]. Zhu et al. [33] believe that China's DC may have had a number of different stages, from the concept of "digital earth" to the establishment and construction of "the national DC geospatial framework" and then from the initial purposes such as to achieve paperless office to the future net platform of the digital China geospatial framework.

Since geographic information-based application systems are stored in different space bases, when the government needs comprehensive information, they cannot be integrated at once, which greatly restricts the progress of urban informatization in the whole coun-

try [34,35]. In such a context, a unified space base and a unified framework are needed to establish the whole city on a map. It is in this environment that the National Administration of Surveying, Mapping, and Geographic Information started the construction of the DC geospatial framework. First, the framework of each province should be unified. Then, a bottom-up strategy should be adopted to gradually establish all the information of the country on a unified spatial basis. Finally, all kinds of information should be communicated on the same spatial basis.

### 2.2. The Main Content of DC Construction in the IoT Environment

Geospatial frameworks will face new opportunities and challenges with the progression of the times, the progress of technology, and the change in human needs. First, the popularity and universality of geographic information is overwhelming, and users are no longer limited to professionals with certain mapping knowledge. It has begun to spread to the masses, and new products that focus on fine 3D quality, panoramic images, and better experience 3D technology are easily accessible to the masses [36]. Second, although the geographic information public platform in DC solves the problem of continuous spatial expression in the real world, it still has a time lag. Chen [37] suggested that with the progression of spatial sensing expansion, such as real-time positioning, IoT, and radio frequency, effective means are provided to integrate real-time information so that the mapping is continuous in the time dimension and can trace back the past, express the present, and predict the future more accurately. Third, with the progression of wireless networks and intelligent terminal technology, machine-centered cities are meant to change into human-centered models. The advent of the mobile office era will help anyone control anything within their jurisdiction in real-time, anytime, and anywhere in the future [38–40]. Fourth, the digital system will no longer adhere to the general query, positioning, statistics, and analysis but will rely on IoT, sensor network, and other in-depth knowledge mining, support scientific decision-making and issue remote control commands to respond to the real world through the analysis and judgment of the digital world [41,42]. Finally, regarding the practical application, the system can make scientific and accurate analyses when ordinary users make requests. Additionally, the computing resources and network resources distributed on each network node can be collected automatically to realize intelligent combinations driven by knowledge and provide services for social needs.

Armstrong et al. [43] suggested that the construction of DC is inseparable from the support of the geospatial framework, which is the responsibility of the Department of Surveying and Mapping Geographic Information as well as the infrastructure of urban informatization. In the DC stage, the main content of the geospatial framework is the basic geographic information database and geographic information public platform. Xu et al. [44] mentioned that with the progression of IoT and cloud computing, basic geographic information databases are transformed into spatiotemporal information databases, and geographic information public platforms are upgraded to spatiotemporal information cloud platforms. The spatiotemporal information database consists of spatiotemporal information data, IoT node address data, a spatiotemporal information database management system, and a support environment. Moreover, a spatiotemporal information cloud platform can provide geographic information, IoT, and node location and develop functional software and interfaces for ubiquitous application environments via visualized fine geographic information and temporal geographic information covering the whole society [45–47]. Compared with geographic information public platforms, spatiotemporal information cloud platforms have better experience, real-time mobility, controllability, and automation. The IoT is a new connection mode established under the Internet of Information and mobile Internet. It originates from the internet and depends on the extension of its boundary and connotation. The IoT changes the situation in which all information on the internet is acquired and created by people, and all items need human instruction and operation, which will profoundly affect every aspect of production and life in the future. In the future, the scale of the interconnection of things in the world will be much larger than the scale

of the interconnection of people. This exponential growth mainly comes from the various connections and autonomous operation between objects. The pattern change from the PC internet to the Internet of Everything is shown in Figure 2. Aba et al. [48] mentioned that the device data of the IoT are generally real-time value data without tampering. Compared with Internet data, data from devices to the IoT platform are transferred to the big data platform, which is a necessary data source for big data applications, unified supervision, and blockchain applications. Sensor-based systems extend our visibility to real-world traffic, utilities, water resources, and buildings, providing new real-time data sources that were previously unavailable or prohibitively expensive to collect [49,50].

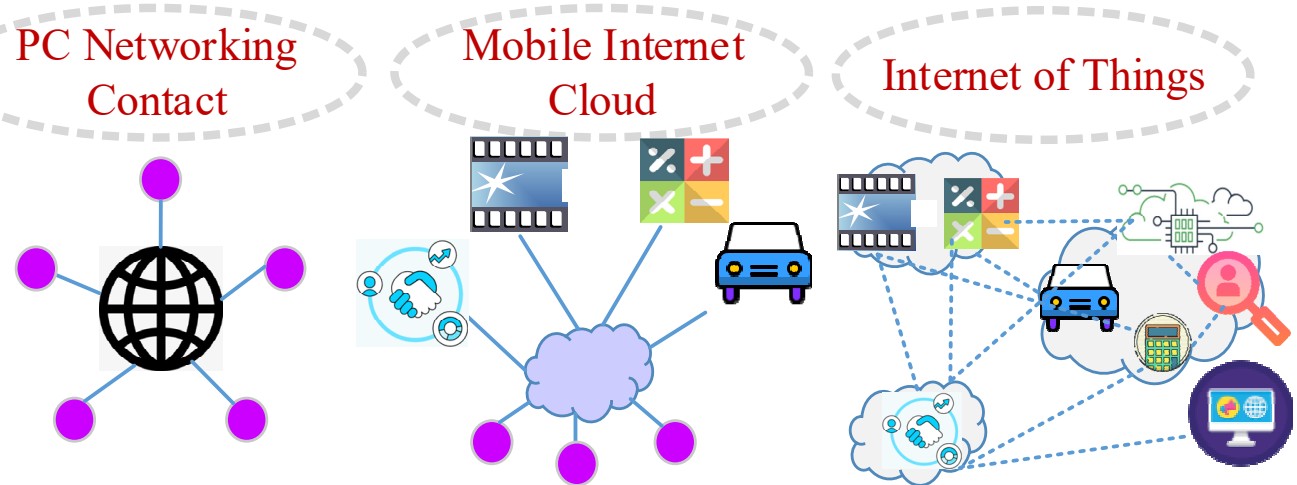

**Figure 2.** Pattern change from PC internet to IoT.

The IoT architecture can be further divided into a perception layer, network layer, platform layer, data analysis layer, and application layer. ElRahman and Alluhaidan [51] suggested that the IoT awareness layer generates and collects data, which is successively transported to the edge side and the center of the platform via 5G communication technology, where edge computing and cloud computing work simultaneously to generate computing power. The data analysis layer performs data preprocessing and analysis by adopting big data technology. AI supports optimization algorithms and ultimately feeds IoT scenarios using cloud computing, data, and the computing power of big data. Various technical elements coexist and depend on each other, and run through the process of data application path. In the future, IoT, 5G, cloud computing, big data, AI, and other technologies will be more closely linked, pushing IoT applications into industrial upgrading and scene intelligence [52,53]. The deconstruction of the IoT's relationship to multiple technologies summarized in this work is shown in Figure 3. In a word, improving technical ability is the key to solving this problem. Only continuous development and breakthroughs in software, hardware, algorithms, and communication technologies can improve the data application capabilities of the IoT. For example, edge computing can release part of the computing power and storage, and the combination with AI can enable rapid responses and decision-making. 5G has obvious effects on reducing delay and improving transmission speed. In addition, the business process system platforms in the public domain are isolated from each other. In practical applications, it is faced with problems such as repeated submission of data, long time limits, and low efficiency. At present, various ministries and commissions are gradually establishing urban operation centers to connect platforms between different business systems to improve the efficiency and security of business and data operations.

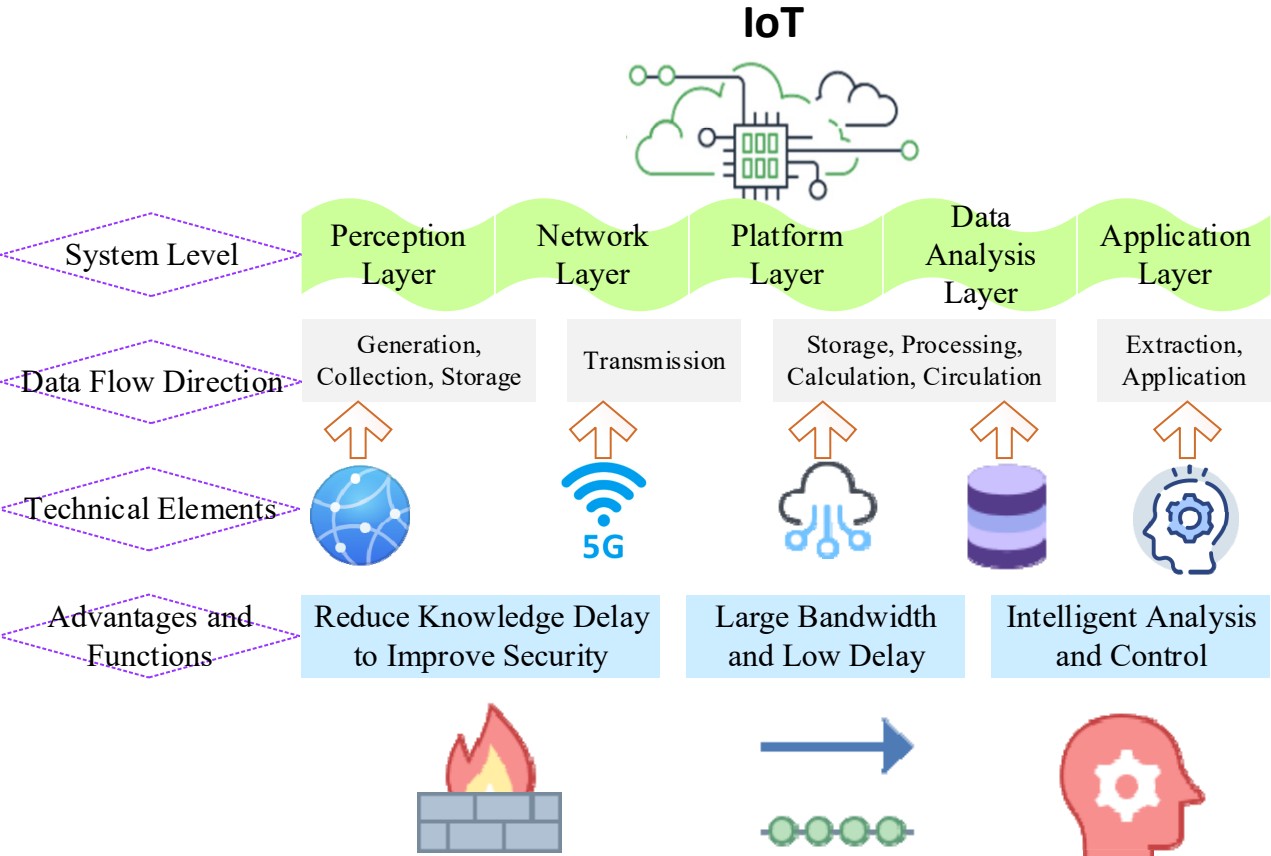

**Figure 3.** A deconstruction of the IoT's relationship to multiple technologies.

### 2.3. DC Management Mode Exploration

DC's management mode achieves accurate and efficient management of urban public utilities relying on modern information technology. Cohen [54] suggested that the core of DC governance realizes the maximization of public interests, and the ultimate goal is promoting the comprehensive development of cities and improving the quality of life of urban residents. The management goal of DC is forming a new social governance paradigm of mutual trust and cooperation among all governance subjects to realize the overall operation across institutions and departments, thereby effectively solving the problems of public management [55,56]. From the perspective of globalization, DC governance has penetrated the administrative systems of all countries in the world, showing a trend of globalization. From a political perspective, information technology changes the way in which citizens participate in public management, thus, enhancing democracy can change the political system. From the administrative level, public institutions in urban management need to connect with government personnel, citizens, and stakeholders to establish balanced performance standards regarding the preferences of different groups. The DC management system embodies innovation in four aspects: (i) the establishment of a city management system that separates supervision from management and achieves the separation of supervision from execution in the management process; (ii) the expansion of effective ways for citizens to participate in urban management by means of SMS, email, service hotline, etc.; (iii) establishment of a sound urban management performance evaluation system to automatically generate real-time evaluation results relying on the information platform as the reward and punishment basis for relevant departments and personnel; (iv) launch the social integrity evaluation system to lay a technical foundation for the implementation of social integrity management.

Gretzel and Koo found that the data center management process of cities in different regions is basically similar on the whole, but there are differences in the specific implemen-

tation [57]. The basic process of DC management includes seven steps: problem discovery, case filing, case dispatch, case disposal, result feedback, audit, and overall evaluation (Figure 4). The DC management system has a professional team of grid supervisors for problem detection. It can find the problems existing in DC management components and events in a timely manner through a regular and thorough inspection and the use of information collection equipment and report the key elements to the city management and supervision center. In the process of filing, problems can be quickly identified, and an accurate judgment can be made by combining the urban event database that has been built and the working standards formulated. If the problem meets the filing criteria, the case will be filed, and the relevant information will be transmitted to the command center through the network. In the case dispatch link, the city management command center sends the case to the administrative agency for disposal through network technology [58,59]. Normally, the district command center can direct the district administration. If the incident is identified as municipal management, it needs to be reported to the municipal command center, and the city will coordinate with the municipal administrative organs for comprehensive disposal. In the case of handling, administrative agencies and professional departments are responsible for specific problems existing in urban management, such as garbage disposal and road damage. In the result feedback link, the disposal department will record the case in detail and complete the disposal of the problem. After the case is disposed, the disposal department will transmit the result report to the command center through the system terminal. In the case review, the grid supervisor checks the site and transmits the results to the monitoring center. If it is determined that the verification results are consistent with the processing feedback information, the case will be closed; otherwise, the monitoring center will resend the case to the command center until the problem is satisfactorily addressed. In the final evaluation process, the DC management system will automatically store all the data generated in the operation process and evaluate the work performance of each department through the assessment of data indicators [60].

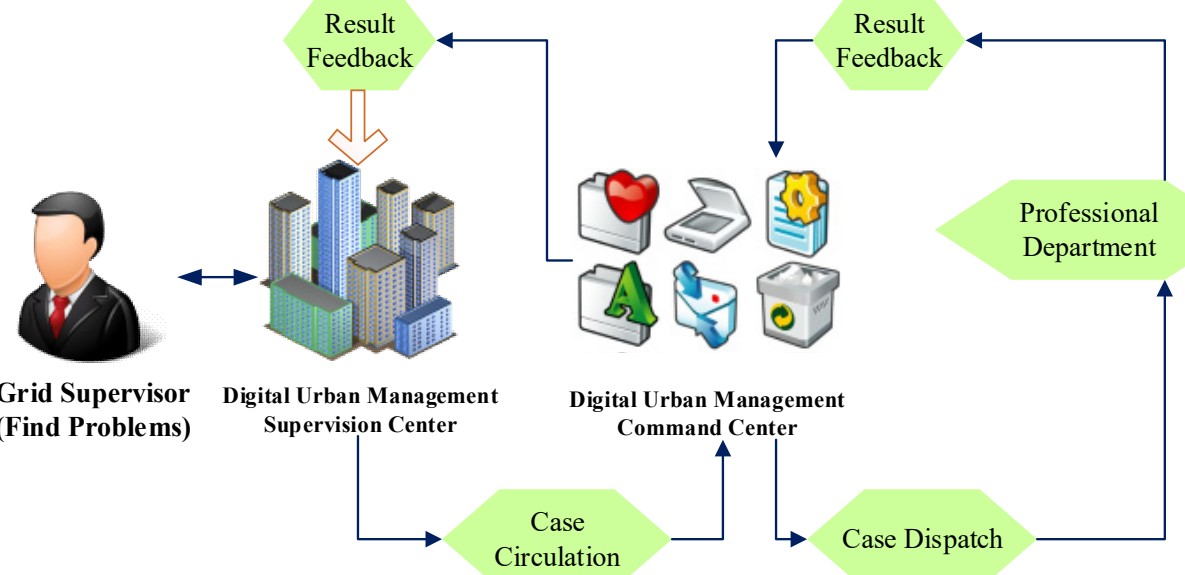

**Figure 4.** The basic flow of DC management.

At present, DC management is mainly achieved through the cell grid management method, code management method, closed-loop management method, and dynamic management method [61–63]. (I) Cell grid management method: first, according to the needs of DC management, the polygon regions with clear boundaries are divided according to the prescribed principles regarding the basic geographic data of large-scale cities. Grid supervisors will be assigned to the cell grid one by one to supervise the individual grid.

(II) The code management method: it can classify urban management components and events according to established standards, assign unique information codes to each type of case, and implement classified management of cases. This digital representation of urban management events in the form of a coding management method can accurately locate and quickly dispose of events, improving the comprehensive efficiency of urban management. (III) Closed-loop management: it has improved the open-loop management mode of traditional urban management, which only focuses on the implementation process and ignores the feedback of results, and established the supervision mode of separating the supervision from the supervision. Through this closed-loop management scheme, it effectively supervises the work of relevant departments of urban management and plays an overall control role in the operation effect of urban management. (IV) Dynamic management: relying on computer information technology, a comprehensive urban management platform with command and dispatch, real-time monitoring, and data analysis is established to realize dynamic monitoring of urban problems. This has changed the traditional extensive and decentralized mode of urban management, made urban management gradually move to a new stage of refinement and intensification, and enhanced the initiative and effectiveness of urban management.

*2.4. A SC Powered by the Metaverse and the DTs*

There are great differences between the metaverse and DTs in industry development, market maturity, and practical application. First, as an unknown concept, the metaverse is still in the state of cognitive cultivation by the government, enterprises, audiences, and other actors. DTs have gone through "concept cultivation", "scheme architecture", "pilot", and "overall implementation construction exploration". Second, in terms of practical applications, the landed DTs city projects focus on transportation, community, construction, and other industrial scenes, covering public service management, community development, and intelligent building applications. The metaverse is in its infancy. Finally, the metaverse is more about building the ideal digital society of shared and immersive experiences. The DTs is more inclined to empower the decision-making of the real society and improve the efficiency of the industry.

The metaverse is a collection of virtual time and space. It is a parallel universe carried by digital form in parallel with human society. Immersive immersion is brought about by augmented reality (AR), virtual reality (VR), and the internet. This may be a milestone in the evolution of digital space, changing the way humans live and the way social space connects with physical space. However, the metaverse is a product form or an idea. In fact, it essentially refers to the form of space Internet connecting time and space in the future after the mobile internet connecting people and the Internet of things connecting things, or a commercial expression of space Internet. A city is an open complex giant system. After many systems interweave and interact with each other, especially after the addition of complex variables such as "people" and "society", it becomes a huge social–physical information system (CPSS). For this kind of system, the description of its running state needs "flow, field, and net" and other systems that are more complex than three-dimensional solid space. The randomness and appearance of multiple magnifications also pose an unbearable challenge to mathematical models trying to discover patterns and predict the future. All that is seen is often the physical form of the city with some highly simplified system states and simulation extrapolations.

It is generally believed that the metaverse is divided into three levels of DTs; digital, primordial, and virtual reality. This also shows that the metaverse is not only accurately simulating the operation of physical space but also creating native experiences in digital space, exploring a unique lifestyle and social form in digital space. In turn, the digital space can also adversely affect the physical space. Therefore, the digital transformation of urban system will shift from DTs to metaverse in terms of thinking mode. For social systems, it is difficult to simulate and calculate comprehensively, no matter the agent model or neural network. By connecting people, citizens and stakeholders can participate in the

decision-making process of city operations in many ways, and the wisdom of people can compensate for the intelligence of machines, which could be a shortcut to making cities truly intelligent.

With the help of the parallel digital system based on meta-cosmic thinking, citizens can participate in the virtual and real interaction and interactive feedback with the urban system at multiple levels, helping the perfection and self-adaptive optimization of the system. This kind of digital system may not be so complicated and difficult to reach technically. It may be the full connectivity experience based on AR, VR, and MR, or the light application entrance, or even web interaction based on a social platform and small program. The meta-cosmic thinking introduces the participation and interaction of multiple subjects under DTs. Thus, the digital platform not only provides visualization ability but also realizes the comprehensive connection between social space and physical space and truly realizes the high degree of cooperation between machine intelligence and human intelligence. In the past decades, the internet has taken over various offline processes, such as shopping and office, from the simple connection between people to the connection between things, realizing the interconnection and intelligent operation of everything. In the new development stage, SC and digital technology are not only facing the comprehensive connection between social space and physical space but also the need to create new experiences, new products, and lifestyles in digital space. At this time, the concept of metaverse emerges. The metaverse should not be a generalized concept blindly hyped, but should identify real industrial trends and respond to them.

## 3. The Main Content and Key Technologies of SC Supported by DTs

### 3.1. Content Architecture and Key Technologies of DTs

According to the adoption of DTs by NASA, DTs creates virtual bodies or digital models equivalent to physical entities [64]. Virtual objects can simulate and analyze physical entities and monitor the running status of physical entities according to the real-time feedback information of physical entities. According to the data of physical entities, the simulation analysis algorithm of virtual entities is improved. This provides more accurate decisions for subsequent operations and improvements of the physical entity [65,66].

DTs are a comprehensive use of perception, computing, modeling, and other information technologies to describe, diagnose, and make decisions on physical space and then realize the interactive mapping of physical space and digital space. It can achieve high-efficiency collaboration, low-cost trial, and intelligent decision-making through data drive, model support, software definition, accurate mapping, and intelligent decision-making. It is suggested that DTs are a dynamic copy of the whole life cycle of entities or logical objects in digital space. It can realize digital representation, simulation tests, and prediction of object state and behavior with high fidelity using rich historical and real-time data and advanced algorithm models. Lee et al. [67] suggested that the effective integration of DTs and lean construction is the key to the success of the project. DTs-based industrial construction has both digital and logistics production lines. The behavior model is first parameterized in the virtual body model establishment, transforming the change in material properties and mechanical properties into the form of parameter change. In addition, the rule model is parameterized. From the perspective of the physical model, the possible parameter changes of behavior and rule level are modularized in the finite element analysis software, and a dynamic DTs library is established. Finally, the whole process is simulated through multilayer model fusion using the DTs library as the driving force.

DTs systems cover six core technologies, namely, perceptual control, data integration, model construction, model interoperation, business integration, and human-computer interaction [68–70]. The first is perceptual control technology, which is capable of data collection and feedback control and is the entrance to the physical world and the outlet of the feedback physical world. The second is data integration technology that can realize the interconnection of heterogeneous devices and systems so that the physical world and the virtual space hosting DTs can seamlessly connect. The third is the model construction

technology to achieve the mapping of physical entity shape and law. The simulation of physical entity shape and known and unknown physical laws is realized through the construction of a geometric model, mechanism model, and data model [71,72]. The fourth is model interoperability technology, which undertakes the task of integrating geometry, mechanism, and data models and realizes the transformation from building a "static mapping physical entity" to building a "dynamic collaborative physical entity". Fifth, business integration technology is the link of DTs value innovation, which can break through the value chain of the whole product life cycle, the whole production process, and the whole business process. The sixth is human-computer interaction technology. Lu et al. [73] believed that workers could feedback control instructions to the physical world through friendly man-machine operation by integrating human factors into the DTs system, thus realizing full closed-loop optimization of DTs.

As the basis of DTs, an algorithm is a new methodology to understand and transform the world. As DTs are more and more closely related to social life and production, the algorithm has a more profound impact on society. Algorithms based on big data and machine deep learning have increasingly strong autonomous learning and decision-making functions. The working principle of a three-dimensional laser scanning system is using the transmitter to transmit laser pulse signals to the target object. The pulse signals are reflected on the surface of the object, and the DTs are received by the laser receiver on the instrument. Finally, the distance between the target object surface and the scanner is calculated according to the time from transmitting to receiving the pulse signals and the propagation speed of the pulse signals. According to the distance between the scanner and the object surface measured by the scanner and the angle information in the horizontal and vertical directions recorded in the scanner, the three-dimensional data of the object surface can be accurately recorded. Finally, these data can be calculated and processed to carry out high-precision point cloud data. Finally, the processed point cloud data can be used to reconstruct the reverse three-dimensional model.

DTs have a wide range of application scenarios in various industries. DTs are innovative applications integrating a series of technologies, such as perception, transmission, computation, modeling, and simulation, and their architecture includes a physical layer, data layer, model layer, and functional layer [74–76]. The physical layer refers to the objects in the physical world, which are divided into tangible objects, such as the human body, objects, and physical space and intangible objects such as business processes. The data layer is the basis of DTs application that consists of data acquisition, transmission, processing, and storage. The model layer is the core of DTs application that adopts modeling and other technologies to represent the digital mirror image of real objects. The function layer is the direct embodiment of the value of DTs. It encapsulates the simulation results and visual services together and provides them to business system applications to satisfy various application scenarios.

### 3.2. Features of DTs in Realizing SC

Since 2020, the outbreak of the novel coronavirus globally has also demonstrated deficiencies in the current urban management and operation of population flow and information connectivity [77–79]. Traditionally, city management is passive. Visualization technology is managed by collecting and processing the events that have occurred in DTs SC entities. With the progression of a new generation of science and technology, such as the fifth generation of communication technology, cloud computing, and edge computing, the progression of informatization has also started a new process [80]. The development of visualization technology has also brought new approaches, and proactive management of cities has become possible [81]. As a visualization technology, according to NASA's authoritative definition, digital town pairing refers to making full use of the physical model, sensor, tag, and other data to integrate the multidisciplinary, multifaceted simulation. In short, DTs is copied in the real world of the physical body, systems, and processes in the network space to form a "clone". The two eventually form a "DTs body".

Deng et al. [82] believed that an important application value of DTs SCs is the interaction and feedback of digital and physical systems in the virtual world and the real world to ensure the coordination and adaptation of physical and digital systems in the whole process of information resource collection. DTs play a crucial role in the field of intelligent manufacturing and effectively promote the effective cooperation between production and design. Some researchers extend the adoption of digital technology from products to workshops and whole enterprises, realizing integration and replication of the total factor, whole process, whole business data, factor management, and production plan [83–85]. These dual digital applications provide valuable inspiration and references for DTs SCs. Through the construction of urban DTs bodies, the interactive operation of the weather environment, infrastructure, population and land, industrial transportation, and other elements can be deduced in the digital world in the form of quantitative and qualitative combinations to draw an "urban portrait". This supports decision-makers in the physical world to achieve the optimal layout of comprehensive benefits of "a map" of urban planning and "a chess game" of urban governance. The DTs+ urban operation management architecture is shown in Figure 5.

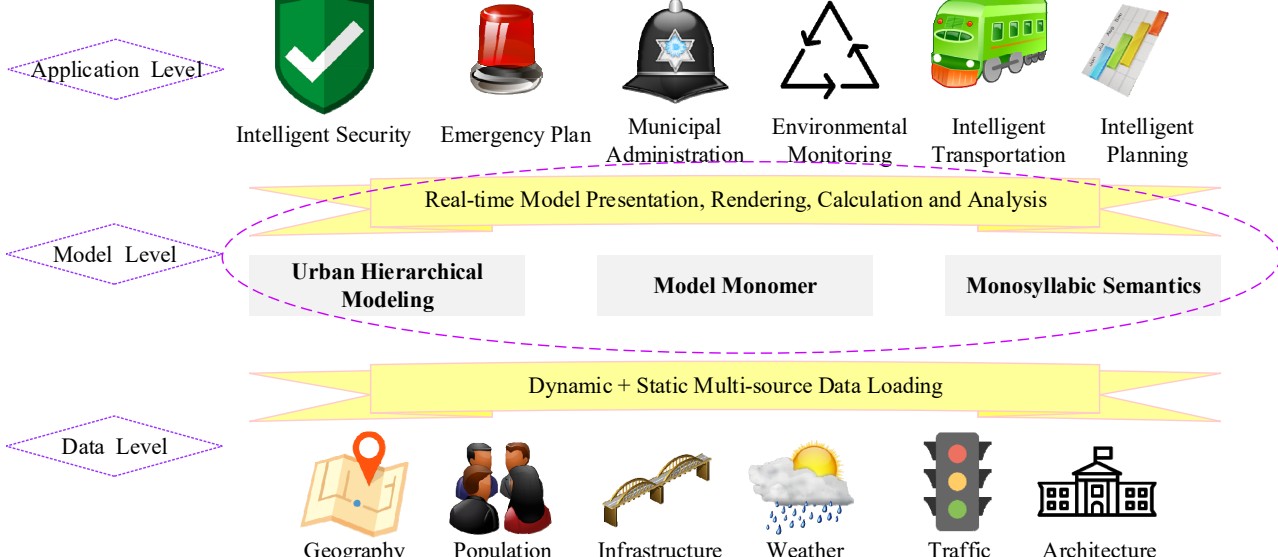

**Figure 5.** DTs+ urban operation management architecture.

DTs urban construction is divided into four stages, namely vision first, IoT perception, application upgrade, and simulation [86,87]. In the first stage, a visual city model is established, which presents static information such as urban infrastructure, buildings, and geographic information in the way of 3D rendering. In the second stage, the hardware devices are gradually increased, such as sensors, cameras, and intelligent terminals, to collect dynamic urban data. In the third stage, technical means are utilized to conduct a basic analysis of the urban operation and make decisions under the DTs urban model platform. The fourth stage is the high-level form of DTs' urban progression. In the case of a comprehensive grasp of the past and present operation data information of the city, the possible state of urban operation is deduced through deep learning and calculation, the solution is designed in advance, and a decision-making simulation is carried out to propose the best solution.

### 3.3. Advantages and Specific Applications of Smart DC

As a large-scale system, the traffic system has received extensive attention in the application scenarios of DTs SCs. However, the traffic system of most SCs is still plagued by problems such as mixed traffic between people and vehicles, roads occupied by motor vehicles, and the slow speed of vehicles. Li et al. [88] indicated that DTs road systems could

be built with the help of the existing IoT foundation, and emergencies on urban roads can be monitored in real-time through the twin road system platform to make reasonable macro-controls to optimize the utilization efficiency of roads and improve the operation efficiency of DTs SCs. DTs traffic can conduct prior simulation evaluation on the scheme in the DTs traffic domain and deliver the real traffic implementation to achieve a purpose. If the target is not achieved, the scheme needs to be repeatedly modified until the expected effect is achieved. On different dates or time periods, different traffic flow loads can be used to adjust the frequency of signal lights and the average traffic flow loads on the road to realize the solution and planning of road problems such as congestion in advance. The parameters of the traffic system in the constructed twin system platform are adjusted to simulate the operation of road traffic, and the traffic parameters are optimized in combination with professional traffic knowledge to improve the performance level of the traffic system [89]. The possible DTs traffic model summarized in this work is shown in Table 1.

**Table 1.** Possible traffic model of DTs traffic.

| Traffic Data Fusion Model | This Model Integrates the Data Perceived by Different Traffic Sensors to Form a Unified Traffic Data Flow, including Traffic Flow and Speed |
|---|---|
| Traffic situation analysis | Based on historical data and real-time traffic perception data, this model analyzes and predicts traffic and evaluates traffic, including traffic flow and speed. |
| Signal control model | According to real-time traffic flow data and prediction of traffic flow, this model optimizes intersection signal timing including single point and route and regional optimal time scheme. |
| Traffic planning model | This model is based on the four-stage method of traffic planning and combines the big data of traffic travel and mobile Internet to predict network traffic volume, including traffic volume and service level. |
| Bus optimization model | Based on the bus network, the model predicts bus travel, evaluates the service level of the bus system, and optimizes the bus network, including the evaluation of the service level of the route optimization. |
| Stop-induced model | Based on the prediction of regional parking space occupancy status and parking demand, the model provides regional parking optimization guidance information, including parking guidance information. |
| Intelligent road model | This model forms a road high-resolution dynamic map according to the real-time perception data of intelligent road and assists connected vehicles to drive safely, including road high-resolution dynamic maps and connected safety tips. |
| Micro simulation model of traffic flow | This model combines vehicle dynamics characteristics and driving and the following model to simulate real-time traffic flow, to compare the advantages and disadvantages of different schemes, including scheme comparison and evaluation. |
| Accident analysis model | This model combines the main factors of accidents to analyze traffic accidents and evaluate road safety, including road safety evaluation and analysis of the main causes of accidents. |
| Auxiliary decision model | This model carries out technical and economic analysis and evaluation of transportation improvement projects and recommends projects with excellent cost performance, including technical and economic analysis and evaluation of projects. |

In the adoption of security warning scenarios, various systems in DTs SCs are intricate, and emergencies in different systems will seriously affect the operation of the city, such as fire, gas leakage, and power supply problems. Traditionally, these events are transmitted through manual notification, which is inefficient and inaccurate in the feedback of events. For early warning and timely response to security incidents, the DTs platform built by sensors and other front-end devices can make timely judgments on emergencies [90,91]. The twin platform provides specific locations and other details of security incidents through sensor sensing and related professional calculations. At the same time, with the aid of AI and other technologies, emergency security incidents can be encountered in the DTs system,

the processing mode of the event can be adjusted, and the response and processing process of the event can be optimized [92]. It can also arrange to rehearse emergency evacuation routes to make the best decisions in a low-cost environment.

DTs visual cities are typically characterized by a global vision, accurate maps, simulations, virtual reality interactions, and intelligent interventions, accelerating the progression of urban governance and application capabilities across sectors [93–95]. Especially in the field of SC governance, it has formed a number of super applications from a global perspective, such as spatial analysis and impact simulation of urban planning, interactive planning and simulation of urban development projects, and urban scenario description, which are used to monitor the normal operation of cities. According to the time and regional development of the urban region, the future development trend is obtained, the law of urban development is understood, and the government is supported to strictly implement it. The urban traffic flow and signal simulation maximize the road capacity. In public services, dual digital analog and three-dimensional interactive experiences will redefine the importance and means of education, health care, and other services [96,97].

Despite some differences, DTs can play an important role in the metaverse. Hypothetically, the hypothetical scenario can be successfully run by providing real data to the components in the meta-universe. Developing construction projects, planning urban areas, and optimizing resource management are just a few of the many uses for DTs in the metauniverse. DTs can help build cities and structures in the metaverse to help visualize how urban areas can be used effectively. The long-term benefits of this application include predicting and managing carbon footprints and suggesting innovative ways to improve the quality of life for city dwellers. The idea of a metaverse helping to optimize city management is supported by technology. The basic technology of DTs has been relatively mature. After the early construction of the Internet of Things and the layout of cameras, offline digital information can be mapped to online, and city dynamics can be mastered in real-time through human-computer interaction. At the same time, urban management will also knock on the door of exploration of the metauniverse through the application of DTs technology. The construction of a 3D base can carry more abundant metaverse application scenarios.

## 4. The Development Prospect and Technological Breakthrough of DC

### 4.1. Problems and Countermeasures in the Construction of New SC

In 2020, the COVID-19 pandemic ravaged the world, causing traditional methods of production and life to face challenges. The demand for informatization and digitalization is surging, making urban digitalization an inevitable trend. In 2021, to realize "dual carbon" and accelerate the pace of high-quality urban progression, digital transformation will become a national strategy and become an important starting point to improve the city's energy level and build core competitiveness. In terms of national strategy, several ministries and commissions have issued policy documents to speed up urban digitalization construction and progression. At present, the tentacles of urban digital transformation are constantly extending to various fields, playing a driving role in improving the level of urban governance, optimizing the quality and efficiency of government services, promoting the innovative development of industries, and facilitating people's living needs. Liu and Chen [98] revealed that it is worth noting that if the government and enterprises are the service providers in the process of urban digital construction, then ordinary people are the witnesses, participants, and beneficiaries of urban digital transformation. Whether people can enjoy the convenience and benefit brought by digitization to work and life is one of the important measurement criteria of digital progression.

There are many issues in the construction of DTs-based SCs. First, there is insufficient ideological understanding. Different regions have different views on SC, and their technical ideas and models are underdeveloped. Second, there is a lack of scientific, reasonable, and systematic top-level design and planning, insufficient prior research and evaluation, emphasis on technical input and neglect of upper structure and mass emotion, and lack

of operation modes for SC construction in different cities. Third, it is difficult to share information on many platforms. There are many independently developed systems by various government departments, and there are information barriers. Fourth, there are loopholes in information security, the government's insufficient emergency response capacity, and the general public's weak awareness of security protection and prevention ability [99,100]. Fifth, there is a shortage of technical innovation talent, senior professionals with professional knowledge, and compound talent who understand government and enterprise management. Sixth, there is a low participation of social forces, great financial pressure in SC construction, and a lack of appropriate mechanisms to attract enterprises and social participation. Seventh, under the influence of the traditional management system, the protection of small interests, the imbalance of progression, and the lag of top-level design, different industries and different fields are developing independently, and the fragmentation of the segmentation is quite prominent, creating many "information islands". This inevitably leads to repeated construction and waste of resources, which also brings great difficulties to the interconnection, integration, and sharing of SC data and information.

To improve the level of SC construction, the following six aspects should be considered. First, it is necessary to strengthen top-level design, formulate scientific and rational plans, formulate medium- and long-term plans for urban progression, strengthen overall planning and guidance, and formulate agreed standards and development models. Second, it is necessary to strengthen scientific and technological innovation, strengthen technological support, strengthen information technology construction, explore the adoption of 5G, big data, and AI in the progression of SCs, make up for the shortcomings of hardware equipment, establish and improve the technical specification system, and strengthen public relations of core technologies. Third, it is necessary to strengthen personnel training and build a contingent of high-quality personnel, introduce high-tech and interdisciplinary talent, form a talent development echelon, and build a contingent of intelligent talent. Fourth, it is necessary to energize the market, boost the drive of industry, give play to the decisive role of the market in resource allocation, innovate ways of government-enterprise cooperation, attract enterprises and design forces to participate, and expand financing channels. Fifth, it is necessary to enhance our awareness of service, improve services for the people, promote the integration of technological innovation and services for the people, and improve product quality and people's sense of experience in light of local conditions and people's needs. Seventh, the government should strengthen top-level design, introduce policies and measures, release unified standards for information, and strengthen guidance and coordination among various departments and industries. It is necessary to breakdown the fragmentation and thoroughly solve the problem of horizontal information "fortress" and vertical information "chimney".

Rathore et al. [101] believed that the foundation of SC is DC. First, the big data center of SC should be established to accumulate and store all the data information of the city, which is the heart of SC. With the help of City Information Modeling (CIM), Building Information Modeling (BIM), Geographic Information Systems (GIS), and other modern means, it is easy to digitize the physical city and then load the data of the urban environment (water quality, air, pollution, etc.) and urban social and economic data (population, industry, assets, etc.) [102–104]. There is also real-time data acquisition, forming a huge city mass "data pool", to achieve the whole field and full coverage of urban informatization and digitization. Second, as the brain of SC, the command center of SC construction and operation should be established. It is responsible for the overall coordination and scheduling of SC construction and operation to form an integrated management and operation pattern with consistent goals, departmental linkage, high coordination, quick response, and efficient operation.

### 4.2. DC 3D Modeling for Fine Management

The progress of digital technology is dependent on the progression of the social economy. The development of the digital age promotes changes in social culture and changes people's living environment to some extent. Through the establishment of a 3D urban

design visualization model, it strives to realize the real-time creation platform of 3D urban design scenes, provides the comparative spatial analysis of urban status quo and urban design, displays the full-size virtual roaming design scheme, and provides planning information analysis and query through the database. In this way, it can realize the creation, editing, and replacement of different architectural models in urban planning, providing a new evaluation method for urban planning evaluation, macro decision-making, and rational construction. Zhang and Kou [105] suggested that the visual city management platform uses virtual reality technology to build a visual three-dimensional virtual simulation system of the city and realizes urban intelligent management with the help of data fusion and interconnection mechanisms.

In SC progression, new digital technologies such as IoT, data analytics, and AI solutions are utilized in different urban verticals to support SC development [106,107]. Dynamic digital technology is combined with static 3D modeling and DTs to expand the city for the understanding of temporal and spatial fluctuations. For the semantic city data model, the global CityGML standard is applied. CityGML is an open and standardized data model for storing and exchanging virtual 3D city models. For the live mesh model, data from existing aerial photographs, point cloud datasets, and laser scans are used. Finally, both the real grid model and the semantic city data model are published as open data for public use. The amount of data will increase more than 100 times compared to existing 2D GIS. Nevertheless, GIS itself is much worse than CAD and 3DMax in 3D data modeling and visualization. Therefore, many scholars have constructed a 3D landscape model of the entire city by combining the advantages of these two systems and planned to use this model to manage the 3D information of the entire city, which was the beginning of the exploration of the 3D city model (3DCM) [108–110]. Using the methods of aerial stereo image pairs and object extraction technology, the semiautomatic measurement of 3D data of aerial image houses is realized, and then the visible surface texture of buildings is restored on the basis of the ground and building surface two-dimensional semi-irregular triangulation network and original digital image, and the 3D urban landscape is reconstructed.

Applying BIM technology to the 3D modeling of DC can simplify the building 3D modeling to the establishment of BIM database, and the 3D model established by using BIM technology is more accurate and more powerful. In addition, there are the following advantages: (i) BIM models are easy to obtain. The BIM model tends to replace architectural design drawings; that is, every new building must establish a BIM model to guide the construction of the project, which provides a strong guarantee for the source of the BIM model. The municipal government can establish special rules and regulations, ordering that new buildings must submit the BIM model to relevant departments for filing. This model can be imported into the 3D city model. (ii) Abundant model information. The BIM model contains abundant information, including building appearance, interior, material, and cost. If necessary, it can be imported into the 3D city model for users to query.

In future DC construction, the integration of BIM data and 3DCM data can promote the construction of 3D urban models. It can not only improve the model quality and schedule but also reduce the modeling cost and shorten the modeling schedule. The adoption of BIM technology in 3D modeling of DC is a new revolution in the progression of DC. A large amount of data sharing and accurate 3D model construction realize seamless information communication between various professional links and provide more efficient and accurate data support for DC management.

### 4.3. Prospect of Digital SC Construction

The internet going digital is only the first step in building a city. The second step is connecting the parts of the city through the DCs of the internet, so that they can be networked. Electronic commerce and electronic government affairs are the network. The third is network communication, and the most basic is the local intelligent response, that is, smart planning stages, smart tolls, smart transportation, etc. The fourth part is the part of the city in the interconnectivity stage. Under the evolution trend of human beings, the

optimization movement aims to realize the SC and basically build an SC. It is an imperative trend to develop from the digitalization of a single system to the deep integration of the whole city system, from system construction to sustainable operation, and from government leading to market-oriented multi-subject participation. In the future, SC will develop from urban digitization to DC, and the whole city will form a "digital giant system" in the digital field. The main vein of urban economic development is from industrial economy and digital economy to intelligent economy. Among them, the digital economy can play a driving role in an industrial economy, thus giving birth to industrial digitization, industrial internet, DTs factories, etc.

First principles thought was first put forward by the Ancient Greek philosopher Aristotle: "In every systematic exploration there are first principles." First principles are fundamental propositions and assumptions that can't be omitted, deleted, or violated. Informationization aims to reduce the trust cost and internal friction in the organization through the efficient circulation of information, thereby achieving diminishing marginal cost. The reduction of the marginal cost of infrastructure and public service is the fundamental driving force of urban agglomeration, which inevitably has internal demand for informatization. With the comprehensive digital transformation of urban planning, construction, operation, and management, cities can adjust their running state at an increasingly high frequency and even in real-time. Urban planning and operation and service links are increasingly inseparable and gradually integrated.

The city's perception system is taken as an example. In the previous SC construction dominated by vertical departments, each department constructed the IoT perception system, such as cameras and sensors. Repeated investments are required from the end equipment to the installation vehicle, and then to the power and network access. In the system construction requirements proposed by each department, they will only consider their own department's business, which will inevitably lead to massive repeated construction, and it also needs to re-gather and integrate various data after decentralized construction. In recent years, the wide application of smart light poles has, to a certain extent, solved the sharing of various roadside terminal installation carriers and power and network access capabilities. Street lamps, signs, 5G base stations, cameras, radars, edge computing nodes, and various roadside sensors can be installed in the same rod. However, further integration of the various cameras installed by multiple departments centralizes the design and overall installation of various sensors. The establishment of an integrated city perception network, as the common foundation of all SC systems, requires a deeper level of top-level design thinking and political wisdom.

SC is a process of long-term operation. More traditional project-based construction is completed without regard to the construction effect, so the development of supply chains is not necessarily superior to the development of long-term operating mechanisms. The new wave ushered in a new phase, and the new phase required a new path. After a systematic analysis of the current social development and technology situation, it is proposed to adopt cloud technology to integrate the letter and create the brain and the overall DC. Based on the construction platform and the way of gathering ecology, the city will evolve together with the digital transformation. As SC construction progresses, governments have realized that they should pay more attention to sustainable operations instead of relying on a single investment in construction. Operation is not a simple IT operation and maintenance. IT needs to rely on market-oriented mechanisms to guide enterprises and citizens to participate in urban innovation and truly solve market pain points. The operation may not be a profitable business model, but it can rely on the value of products and services to make reasonable fees and achieve sustainable and healthy development.

## 5. Discussion

DTs connect the physical world with the virtual space and create the real world in the virtual space by using historical data, real-time data, and algorithmic models. This technology will greatly promote the comprehensive situation analysis of factories, equip-

ment operation and maintenance, production line operation, production management, and other changes and will become the technical basis and development trend of intelligent manufacturing, "Industry 4.0". From the construction of the "urban brain" and SC to the promotion of the "digital transformation" of cities and the recently discussed DTs cities, digitalization has not only been a concern of science and technology circles and entrepreneurs but has also become a topic for urban administrators of all countries. DTs SC can integrate the fragmented information of the IoT, solve all the needs and pain points at one site, and build a new map of the IoT industry in the form of a metaverse.

The digital wave represented by the Internet of Things, big data, artificial intelligence, and other new technologies have swept the world. The physical world and the information world are developing in parallel and interacting, and the construction of DTs cities has become unstoppable. The development of DTs SC needs good urban planning, but various urban problems caused by unreasonable urban planning still exist significantly. DTs visualization technology can make things that cost relatively high in reality and are difficult to achieve through short-term experiments quickly, low cost, and multiple implementations in virtual entities. The model construction of various schemes on the virtual platform can be used to guide the construction in reality, which can be used to deduce urban planning. By setting spatial and event thresholds, it can be judged whether the city is different from the planned goals in its development. Therefore, in the application of urban planning, it is necessary to record the construction phase of the full cycle and operate after the construction is completed. For the application of DTs technology, it is necessary to record all elements in the dimensions of space and time. Through the construction of the DTs city platform, the perception and decision-making ability of the city can be improved, and future planning and development can also bring a broader vision. Under the premise that all social subjects can have access to the DTs platform, the DTs technology creates a collaborative environment of multiple cooperation and co-governance for all subjects of urban risk management, which can improve the social identity and the sense of belonging of the public, and improve their enthusiasm to participate in urban risk management.

## 6. Conclusions

This work focuses on the discussion of digitalization and the impact of DTs on the progression of modern cities, plus the main contents and key technologies of SCs supported by DTs. A DTs city is an innovative application under the integration of a series of technologies, such as perception, transmission, computing, modeling, and simulation. Urban construction is divided into four stages of vision first: IoT perception, application upgrade, and simulation. The construction of an SC based on DTs improves the management and allocation efficiency of urban public resources and provides decision-makers with a tool to perceive the overall situation of the city and a platform for command and scheduling. This work is of important reference value for the subsequent innovation and development of DTs cities, but it fails to further discuss the hardware and software components in the DTs modeling process, which will be gradually improved in subsequent research.

**Author Contributions:** Conceptualization, Z.L. and M.G.; methodology, W.-L.S.; investigation, Z.L.; writing—original draft preparation, Z.L.; writing—review and editing, W.-L.S.; supervision, M.G. All authors have read and agreed to the published version of the manuscript.

**Funding:** This research received no external funding.

**Institutional Review Board Statement:** Not applicable.

**Informed Consent Statement:** Not applicable.

**Acknowledgments:** Thanks for the reviewers' suggestions.

**Conflicts of Interest:** The authors declare no conflict of interest.

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
