# Peer review of "Impact of Digital Twins and Metaverse on Cities: History, Current Situation, and Application Perspectives"

_applsci, doi:10.3390/app122412820_

Round 1
Reviewer 1 Report
Dear Editor and Authors, follow the manuscript review:
Impact of Digital Twins and Metaverse on Cities: History, Current Situation, and Application Perspectives
The topic is relevant and of great interest to the academic community. However, the general logic of the argumentative construction throughout the article is difficult to follow. The current form of the paper makes it very descriptive and not very analytical.
I suggest some changes to the authors:
1) The abstract should clearly show the methodological procedures adopted in conducting the research.
2) Since the Introduction, the authors should explore in depth the relationship between DT and the Metaverse. In addition, authors should state the main contributions of the paper at the end of section 1 (Introduction). This will help the readers and will also enhance the value of the paper.
3) I suggest that the authors develop a section explaining the methodological procedures adopted to develop the paper, seeking linkage between the aspects covered in sections 2, 3 and 4.
4) Subsequently, the topics covered in sections 2, 3, and 4 can be presented in a more analytical and less descriptive way.
5) I also suggest authors to elaborate a specific discussion section in the article
Author Response
The topic is relevant and of great interest to the academic community. However, the general logic of the argumentative construction throughout the article is difficult to follow. The current form of the paper makes it very descriptive and not very analytical.
I suggest some changes to the authors:
1) The abstract should clearly show the methodological procedures adopted in conducting the research.
Reply:Thanks for the suggestion. We have improved the abstract to supplement the methods and procedures of this study. Combining the technical elements of digital twinning, this paper discusses the coupling effect of digital twins technology and urban construction, and the internal logic of digital twins technology embedded in urban construction.
2) Since the Introduction, the authors should explore in depth the relationship between DT and the Metaverse. In addition, authors should state the main contributions of the paper at the end of section 1 (Introduction). This will help the readers and will also enhance the value of the paper.
Reply:Thanks for the careful reading. We have added the analysis of the relationship between DT and Metaverse in the second paragraph of the introduction section. From the perspective of interaction, Metaverse focuses on augmented reality technology to build a virtual world beyond the real world in order to obtain an immersive user experience, while Digital Twins focuses more on virtual reality technology to generate the image of the real world. At the end of the introduction section, the main contributions of this paper are explained. The construction of a smart city based on digital twins improves the management and allocation efficiency of urban public resources, and provides decision makers with a perception tool of the overall situation of the city and a platform for command and dispatch.
3) I suggest that the authors develop a section explaining the methodological procedures adopted to develop the paper, seeking linkage between the aspects covered in sections 2, 3 and 4.
Reply:We have supplemented section 1.2, summarized the methods and procedures of this study, and connected the contents of sections 2, 3 and 4 in series. In section 2, the historical process and construction conception of digital city under the demand of modernization are defined. In section 3, the key technologies and functions of DTs in smart city construction are explored. In section 4, the development prospect and technological breakthrough of digital smart city are comprehensively planned.
4) Subsequently, the topics covered in sections 2, 3, and 4 can be presented in a more analytical and less descriptive way.
Reply:Thanks for the careful reading. We have streamlined some of the contents covered in sections 2, 3 and 4, and deleted some redundant details.
5) I also suggest authors to elaborate a specific discussion section in the article
Reply:Thanks for the suggestion. We have added the discussion section, section 5, to give an overall overview and analysis of the full text. For the application of digital twins technology, it is necessary to record all elements in the dimension of time and space. By building a digital twins city platform, the perception and decision-making ability of the city can be improved, and it will also bring a broader vision to the future planning and development.
Reviewer 2 Report
I thank the authors for submitting their manuscript entitled “Impact of Digital Twins and Metaverse on Cities: History, Current Situation, and Application Perspectives”. This article addresses a timely topic by investigating how digital twins can be applied in smart cities. However, I have major concerns with this article in its current form. Before it can be published in an outlet like Applied Science, the authors are adviced to implement substantial changes to improve their manuscript’s quality.
First, it is glaring that no details regarding the conducted research method are included. One can only guess what the authors did to come to their findings. Did they conduct interviews, field experiments, a literature review? Although I assume a literature review was conducted, details about the literature selection process are necessary. The authors are therefore advised to add a method section or detailed explanations about their research process and method.
Furthermore, it seems like smart city and digital city are used interchangeably throughout the paper. Please be precise here and stick to one term or explicitly indicate that you use these terms synonymously.
Furthermore, the article is merely about the potential of digital twins. Although the article’s title indicates that metaverses are investigated, too, this only seems to be partly the case. If the potential fo metaverse is really part of this study, authors should elaborate the findings on this technology. In its current form, I would recommend to sharpen the paper and to deal only with digital twins.
Finally, there are typos and grammatical errors. I would therefore recommend to undergo a proofreading before the article is published.
Although my concerns are substantial, I believe these can be addressed in a revised version. I therefore would recommend major revisions and wish the authors good luck with improving their manuscript.
Author Response
I thank the authors for submitting their manuscript entitled “Impact of Digital Twins and Metaverse on Cities: History, Current Situation, and Application Perspectives”. This article addresses a timely topic by investigating how digital twins can be applied in smart cities. However, I have major concerns with this article in its current form. Before it can be published in an outlet like Applied Science, the authors are adviced to implement substantial changes to improve their manuscript’s quality.
First, it is glaring that no details regarding the conducted research method are included. One can only guess what the authors did to come to their findings. Did they conduct interviews, field experiments, a literature review? Although I assume a literature review was conducted, details about the literature selection process are necessary. The authors are therefore advised to add a method section or detailed explanations about their research process and method.
Reply:Thanks for the careful reading. We have explained the method of literature review in the abstract. We collected, evaluated and sorted out the related literature of omni-directional coverage digital twins technology and its application, connected the related research in series, and summarized the relevant recognized conclusions according to the modules.
Furthermore, it seems like smart city and digital city are used interchangeably throughout the paper. Please be precise here and stick to one term or explicitly indicate that you use these terms synonymously.
Reply:Thanks for the careful reading. There is a difference between smart city and digital city. This article needs to explain it in different terms. Digital cities pay attention to the production, accumulation, and application of data resources, while smart cities pay more attention to service design and provision from the perspective of users. In addition, while digital cities are committed to realizing various functions of urban operation and development by means of information technology, smart cities put more emphasis on shaping urban public value through the participation of government, market, and social forces.
Furthermore, the article is merely about the potential of digital twins. Although the article’s title indicates that metaverses are investigated, too, this only seems to be partly the case. If the potential fo metaverse is really part of this study, authors should elaborate the findings on this technology. In its current form, I would recommend to sharpen the paper and to deal only with digital twins.
Reply:Thanks for the suggestion. We have introduced the connection between digital twins and metaverse and its application in the realization of metaverse in the introduction and section 3.3. The idea that the metaverse helps to optimize urban management is supported by technology. The basic technology of digital twins has been relatively mature. After the construction of the Internet of Things and the arrangement of cameras in the early stage, offline digital information can be mapped to online, and the city dynamics can be grasped in real time through human-computer interaction.
Finally, there are typos and grammatical errors. I would therefore recommend to undergo a proofreading before the article is published.
Reply:That is a good suggestion! We have checked the spelling and grammar in the text and corrected the mistakes to ensure the high quality of the language.
Although my concerns are substantial, I believe these can be addressed in a revised version. I therefore would recommend major revisions and wish the authors good luck with improving their manuscript.
Reply:Thanks for the careful reading. We have revised your suggestions one by one, hoping to improve the overall quality of the paper.
Reviewer 3 Report
1.Introduction part is too generic and it should be problem specific.
2. Literature related to the Management of Direct Current could be presented in terms of deriving into multiple layers and
the algorithms relavant to the problem addressed could be narrated.
3. Too much of descriptive and title could be as case study/review on the construction of DC Models.
4. The introduction has described in terms of networking, IoT Layer and application layer, Even in title it is mentioned as Metaverse
and justify the relavance with this article.
5. Title of the paper to be narrated according to the study done.
Author Response
1.Introduction part is too generic and it should be problem specific.
Reply:Thanks for the suggestion. We have improved the introduction and divided it into two subsections, namely the research background, research methods, procedures and significance.
2. Literature related to the Management of Direct Current could be presented in terms of deriving into multiple layers and the algorithms relavant to the problem addressed could be narrated.
Reply: That is a good suggestion! In section 3, we mainly described the current situation of smart cities supported by digital twins. Now, we have supplemented the paragraphs of coherent logic, promoted the progressive level of related literature, and explained the key algorithms. With the closer connection between digital twins and social life and production, the impact of algorithm on society is more profound.
- Too much of descriptive and title could be as case study/review on the construction of DC Models.
Reply: That is a good suggestion! We have appropriately deleted the introduction of smart cities, and changed the title of section 4.
4. The introduction has described in terms of networking, IoT Layer and application layer, Even in title it is mentioned as Metaverse and justify the relavance with this article.
Reply: Thanks for the suggestion. We have supplemented the correlation between digital twins and metaverse in the introduction and section 3.3. The idea that the metaverse helps to optimize urban management is supported by technology. - Title of the paper to be narrated according to the study done.
Reply: Thanks for the careful reading. We have made sure that the information covered in the title of the paper is reflected in the text.
Round 2
Reviewer 1 Report
I recomend accept this paper
Reviewer 2 Report
I thank the authors for submitting the revised manuscript. The suggested changes have been applied carefully and in my view, the paper can be accepted in its current form.